

# Airborne ultra-wideband radar sounding over the shear margins and along flow lines at the onset region of the Northeast Greenland Ice Stream

Steven Franke[1], Daniela Jansen[1], Tobias Binder[1,*], John D. Paden[2], Nils Dörr[3], Tamara A. Gerber[4], Heinrich Miller[1], Dorthe Dahl-Jensen[4], Veit Helm[1], Daniel Steinhage[1], Ilka Weikusat[1,5], Frank Wilhelms[1,6], and Olaf Eisen[1,7]

[1]Alfred Wegener Institute, Helmholz Centre for Polar and Marine Sciences, Bremerhaven, Germany
[2]Centre for Remote Sensing of Ice Sheets (CReSIS), University of Kansas, Lawrence, KS, USA
[3]Institute of Photogrammetry and Remote Sensing, Karlsruhe Institute of Technology, Karlsruhe, Germany
[4]Physics of Ice, Climate, and Earth, Niels Bohr Institute, University of Copenhagen, Denmark
[5]Department of Geosciences, Eberhard Karls University Tübingen, Germany
[6]Department of Crystallography, Geoscience Centre, University of Göttingen, Germany
[7]Department of Geosciences, University of Bremen, Bremen, Germany
[*]now at Ibeo Automotive Systems, Hamburg, Germany

**Correspondence:** Olaf Eisen (olaf.eisen@awi.de) and Daniela Jansen (daniela.jansen@awi.de)

**Abstract.** We present a high-resolution airborne radar data set (EGRIP-NOR-2018) for the onset region of the Northeast Greenland Ice Stream (NEGIS). The radar data were acquired in May 2018 with Alfred Wegener Institute's multichannel ultra-wideband (UWB) radar mounted on the Polar6 aircraft. Radar profiles cover an area of $\sim 24\,000$ km$^2$ and extend over the well-defined shear margins of the NEGIS. The survey area is centred at the location of the drill site of the East Greenland Ice-Core Project (EastGRIP) and several radar lines intersect at this location. The survey layout was designed to: (i) map the stratigraphic signature of the shear margins with radar profiles aligned perpendicular to ice flow, (ii) trace the radar stratigraphy along several flow lines and (iii) provide spatial coverage of ice thickness and basal properties. While we are able to resolve radar reflections in the deep stratigraphy, we can not fully resolve the steeply inclined reflections at the tightly folded shear margins in the lower part of the ice column. The NEGIS is causing the most significant discrepancies between numerically modelled and observed ice surface velocities. Given the high likelihood of future climate and ocean warming, this extensive data set of new high-resolution radar data in combination with the EastGRIP ice core will be a key contribution to understand the past and future dynamics of the NEGIS. The EGRIP-NOR-2018 radar data products can be obtained at the PANGAEA Data Publisher (https://doi.pangaea.de/10.1594/PANGAEA.928569; Franke et al. 2021a).





## 1 Introduction

The Northeast Greenland Ice Stream (NEGIS) efficiently drains a large area of the Greenland Ice Sheet and is a crucial component of the ice sheet mass balance (Fahnestock et al., 1993; Rignot and Mouginot, 2012). It extends from the central ice divide over more than 600 km towards the northeastern coast, where it discharges ice through the three marine-terminating glaciers (79 N Glacier, Zachariae Isbræ and Storstrømmen Glacier). The currently prevailing hypothesis is that an anomaly of elevated geothermal heat flux (GHF) leads to extensive basal melting (Fahnestock et al., 2001) and induces ice flow. The GHF at the onset of NEGIS is the most important and, at the same time, most uncertain parameter in the representation of basal melt and the subglacial hydrology in ice flow models (Smith-Johnsen et al., 2020). However, the large gap between observed and modelled basal melt rates (Gerber et al., 2021; Zeising and Humbert, 2021; MacGregor et al., 2016; Buchardt and Dahl-Jensen, 2007) inside the ice stream and in the broader surrounding of the NEGIS onset, raise questions about the real thermal state at the bed. The exceptionally high GHF proposed by Fahnestock et al. (2001) would be non-compatible with known geological processes (Jóhannesson et al., 2020; Blackwell and Richards, 2004). Therefore, Bons et al. (2020) raise the question of how fast flow at the NEGIS onset is possible without an extraordinary high basal heat flux and meltrates.

Unlike other ice streams in Greenland, the NEGIS lacks an extensive overdeepened bed, and thus lateral topographic constraints (Joughin et al., 2001; Franke et al., 2020). The ice stream more or less symmetrically broadens along flow, as more ice is dragged through the shear margins (Fahnestock et al., 2001; Joughin et al., 2001). Subglacial water routing in combination with subglacial till deformation seems to be a further controlling mechanism of ice flow at the onset of the NEGIS (Keisling et al., 2014; Christianson et al., 2014). The outlet area is characterised by an overdeepened basin covered with unconsolidated sediments (Joughin et al., 2001; Bamber et al., 2013). Ice thinning as a consequence of increasing oceanic water temperatures around Greenland, could thus potentially be transmitted far upstream (Christianson et al., 2014), and changes in the hydropotential might have significant effects on the ice stream geometry. The high susceptibility of NEGIS to marine-triggered discharge and the expected increase in ocean water temperatures in the years to come (Yin et al., 2011; Straneo et al., 2012) raises questions about the future ice stream stability and its effect on the Greenland Ice Sheet mass balance.

Large scale ice flow models are essential tools to predict the future behaviour of glaciers and ice sheets and are necessary to estimate future sea-level rise. Until today, these models fail to successfully simulate the NEGIS due to insufficient understanding of the key processes responsible for ice flow dynamics (Mottram et al., 2019; Shepherd et al., 2020), limiting the prediction accuracy. The East Greenland Ice-Core project (EastGRIP; https://eastgrip.org) aims to drill a deep ice core in the upstream area of the NEGIS, providing valuable insights into the climate record, basal properties and ice flow history at the drill site. Ice cores provide *in situ* information on physical and chemical properties at high resolution but are limited as being a spatial point measurement. Hence, further geophysical techniques are required to extrapolate this information to obtain a complete picture of the ice dynamic properties.

Radio-echo sounding (RES) has long become a standard method in glaciology. The polar ice sheets as well as low-latitude glaciers and ice caps have extensively been covered by airborne (e.g. Steinhage et al., 1999; Schroeder et al., 2020) or ground-based (e.g. Pälli et al., 2002) RES surveys. The transmitted electromagnetic waves are sensitive to changes in dielectric per-





mittivity and electrical conductivity and get reflected, scattered or refracted at interfaces of dielectric contrasts in the medium they propagate (Fujita et al., 1999). The most common glaciological application is the sounding of ice thickness and bedrock topography (e.g. Hempel and Thyssen, 1992; Dahl-Jensen et al., 1997; Steinhage et al., 1999; Nixdorf and Göktas, 2001; Kanagaratnam et al., 2001). Reflections within the ice column, or so-called internal reflection horizons (IRH), are often caused by impurity layers of volcanic origin representing isochronous horizons (Millar, 1981). These provide valuable information on the ice flow regime and strain history (Vaughan et al., 1999; Jacobel et al., 1993; Hodgkins et al., 2000) and can be used to reconstruct past accumulation rates (e.g. Richardson et al., 1997; Nereson et al., 2000; Siegert and Hodgkins, 2000; Pälli et al., 2002; Nereson and Raymond, 2001), to match ice cores from different locations (e.g. Jacobel and Hodge, 1995; Siegert et al., 1998; Hempel et al., 2000) and to validate numerical ice flow models (e.g. Huybrechts et al., 2000; Baldwin et al., 2003). Further applications of RES include the detection of crevasses (e.g. Zamora et al., 2007; Eder et al., 2008; Williams et al., 2014), mapping of subglacial lakes and basal hydrology (e.g. Carter et al., 2007; Palmer et al., 2013; Young et al., 2016), identifying thermal regimes (e.g. Murray et al., 2000; Copland and Sharp, 2001), determining snow and firn genesis (e.g. Frezzotti et al., 2002) and obtaining information on the crystal orientation fabric (e.g. Matsuoka et al., 2003; Eisen et al., 2007; Jordan et al., 2020).

We present unique airborne radar data of the onset region of NEGIS recorded in 2018 by a multichannel ultra-wideband radar system. The data set consists of profiles oriented parallel and perpendicular to the ice flow direction. The high along-track and range resolution allows consistent isochrone tracing, providing insights into the three-dimensional structure of the ice stream. In combination with the EastGRIP ice core, this dataset contributes to a better understanding of ice flow dynamics of the NEGIS. In our manuscript, we introduce the study site and survey design. Furthermore, we describe the radar data processing and the respective data products. The data are freely available at the PANGAEA data publisher (https://doi.pangaea.de/10.1594/PANGAEA.928569).

## 2 Survey region and previous work

In May 2018, we recorded radar data in the vicinity of the drill site of the EastGRIP ice core. An area of ~ 24,000 km$^2$ was mapped with 7494 km of radar profiles along flow lines and perpendicular to ice flow (Figure 1). The survey region extends ~ 150 km upstream and downstream of the EastGRIP drill site and ranges from the central part of the ice stream up to 50 km beyond the shear margins. In our survey region, the ice stream accelerates from ~ 10 to more than 80 m a$^{-1}$ and widens from ~15 to ~55 km. The radar data also covers the transition in the position of the shear margin as well as strongly folded internal stratigraphy outside of the ice stream (see Figure 1). Figure 1b and c show the locations of radar profiles in relation to the ice surface velocity. Profiles extending perpendicular to ice flow have a spacing of 5 km in the region close to the drill site and 10 km further up- and downstream. Along-flow profiles either follow flowlines, which in some cases pass through the shear margins, or are constantly located inside the ice stream. Other profiles are oriented parallel to ice flow of the NEGIS but are located outside of the ice stream.





The ice thickness in our survey region ranges from 2059 m to 3092 m and shows, on average, a gradual decrease in thickness from the upstream to the downstream part (Franke et al., 2020, 2019). An analysis of the bed topography, basal roughness and bed return echoes by Franke et al. (2021b) shows that our survey area can be divided into two different morphological regimes. The upstream part (upstream of EastGRIP) is characterised by a narrow ice stream width with accelerating ice flow

velocity, a smooth bed with elongated flow-parallel subglacial landforms and a soft till layer at the base (Christianson et al., 2014). Downstream of EastGRIP, the ice stream widens, and we note an overall change to a rougher and more variable bed geomorphology. The ice stream widens up to 57 km and ice flow velocity keeps constant and decreases locally (Franke et al., 2021b).

In the 2012 summer field season, a scientific consortium collected ground-based geophysical data (RES and seismic survey)

as well as a shallow ice core (Vallelonga et al., 2014). Christianson et al. (2014) examined the ice-bed interface by means of radar and seismic data analysis. They found high-porosity, water-saturated till, which lubricates the ice stream base and most likely facilitates ice stream flow. Keisling et al. (2014) used the same radar data to analyse the internal radar stratigraphy and suggest that the basal hydrology controls the upstream portion of the NEGIS. By contrast, the downstream part is rather confined by the bed topography. Riverman et al. (2019a) and Riverman et al. (2019b) analysed the shear margins in particular

and observed an increased accumulation and enhanced firn densification in the upper ice column as well as wet elongated subglacial landforms at the bed. Furthermore Holschuh et al. (2019) use RES data to evaluate 3-D thermomechanical models of the NEGIS. The authors highlight the complexity of the stagnant to streaming ice flow transition and provide insights into the englacial heat transport. Finally, a comprehensive chemical analysis of a 67 m deep firn core was conducted by Vallelonga et al. (2014). The results demonstrated that a deep ice core at this location has the potential to retrieve a reliable record of the

Holocene and last-glacial cycle.

## 3 Methods

### 3.1 Radar data acquisition

The ultra-wideband (UWB) airborne radar is a Multichannel Coherent Radar Depth Sounder (MCoRDS, version 5) which was developed at the centre for Remote Sensing of Ice Sheets (CReSIS) at the University of Kansas (Hale et al., 2016). It has

an improved hardware design compared to predecessor radar depth sounders by CReSIS (Gogineni et al., 1998; Wang et al., 2015). The radar configuration deployed in 2018 consists of an eight-element radar array mounted under the Polar 6 Basler BT-67 aircraft's fuselage. The eight antenna elements function as transmit and receive channels using a transmit-receive switch. The total transmit power is 6 kW, the radar can be operated within the frequency band of 150 – 600 MHz. The pulse repetition frequency (PRF) is 10 kHz, and the sampling frequency is 1.6 GHz. The characteristics of the transmitted waveform as well as

the recording settings can be manually adjusted. We refer to the combined transmission/reception settings as *waveforms* in the following.

All profiles were recorded using linear frequency-modulated chirps in the frequency band of 180-210 MHz, antenna elements oriented with the E-plane aligned with the along-track (HH polarisation), and the transmit antenna beam pointed toward nadir.

**Figure 1.** Location of the EGRPI-NOR-2018 survey area in Northeast Greenland with the MEaSUREs ice surface velocity from Joughin et al. (2017) in the background. (a) The radar survey lines are centred at the EastGRIP drill site (white triangle) and extend up to 150 km upstream and downstream of the NEGIS. The locations of one along-flow radar section (Figure 2; blue) and three cross-flow radar sections (Figure 3; red) are shown in panel (a). The ice flow direction of the ice stream is indicated with a white arrow. 3D images of the cross-flow profiles and along-flow profiles are shown in (b) and (c), respectively. The radar sections are shown with a vertical exaggeration factor of z = 10. The ice surface velocity (Joughin et al., 2017) is shown on a logarithmic scale (log10) for entire Greenland (upper left image) and on a linear scale for panels (a), (b) and (c). For panel (b) and (c) the ice surface velocity is projected on top of the bed topography model from Franke et al. (2020). The projection for all maps is: WGS 84 / NSIDC Sea Ice Polar Stereographic North (EPSG:3413).



**Table 1.** Acquisition parameters of the EGRIP-NOR-2018 radar campaign.

| Parameter | Value |
|---|---|
| Radar system | MCoRDS5 |
| Frequency range | 180-210 MHz |
| Waveform signal | $1\mu s$, $3\mu s$, $10\mu s$ chirp |
| Waveform presums[a] | 2, 4, 32 |
| Pulse Repetition Frequency | 10 kHz |
| Sampling frequency | 1600 MHz |
| Tukey window taper ratio | 0.08 |
| Transmit channels | 8 |
| Receiving channels | 8 |
| Aircraft altitude above ground | $\sim 360\,\mathrm{m}$ |
| Aircraft velocity | $\sim 260\,\mathrm{km\,h^{-1}}$ |

[a]Presums are set for each waveform individually.

We used three alternating waveforms to increase the dynamic range of the system (see Table 1). Short pulses (1 $\mu$s) and low
receiver gain of 11 dB to image the glacier surface, and longer pulses (3 - 10 $\mu$s) with higher receiver gain (48 dB) to image
internal features and the ice base. Recorded traces were coherently presummed with zero-pi modulation in the hardware (Allen
et al., 2005) to reduce the data rate and to increase signal-to-noise ratio (SNR), leading to a reduced effective PRF. The presum
factors were selected with regard to the pulse length of the respective waveform. To reduce range sidelobes without losing
much signal power, the transmitted and the pulse compression filter were amplitude-tapered using a Tukey window with a
taper ratio of 0.08 (Li et al., 2013).

Before the data acquisition, the amplitude, phase and time delay of the antenna elements were equalized during a test flight
over open water during the transit to Greenland. During data acquisition, the position of the aircraft was determined by four
NovAtel DL-V3 GPS receivers, which are sampling at 20 Hz. The GPS system operates with dual-frequency tracking so that
the position accuracy can be enhanced during post-processing.

## 3.2   Radar data processing

The acquired data comprised 24 total radargrams, one from each pairwise combination of 8 receivers and 3 waveforms. The
post-flight processing goal was to create single radargrams of the profiles covering the ice sheet from surface to bed with high
SNR, fine resolution and high dynamic range. The main processing included pulse compression in the range dimension, syn-
thetic aperture radar focusing in the along-track dimension and array processing in the cross-track/elevation-angle dimension.
Lastly, we vertically concatenated the radargrams of the three waveforms. The post-processing tools are implemented in the
CReSIS Toolbox (CReSIS, 2020b).





At first, the recorded traces were pulse compressed using a Tukey time domain weighting on the pulse and frequency domain matched filtering with a Hanning window to reduce sidelobes. For this purpose, the matched filter duplicates the transmitted waveforms based on the radar transmit settings.

SAR processing was carried out to focus the SAR radargrams in the along-track direction. The SAR processing is based on the fk-migration technique for layered media (Gazdag, 1978), which was adapted for radioglaciology (Leuschen et al., 2000). We used a two-layered velocity model with constant permittivity values for air ($\epsilon_r = 1$) and ice ($\epsilon_r = 3.15$). The air-ice interface was tracked using quicklook imagery, which is generated using 20 coherent averages followed by five incoherent averages by an automated threshold tracker. Platform motion compensation is applied during averaging. The SAR aperture

length at each pixel was chosen to create a fixed along-track resolution of 2.5 m. A requirement for the fk-migration is a uniformly sampled linear trajectory of the receivers along the SAR aperture extent. Changes in aircraft elevation, roll, pitch and heading lead to phase errors in the migrated data, thus to decreased SNR and blurring. Processed GPS and INS data in high precision from the aircraft were used to correct these effects. The motion compensation consisted of (1) uniform resampling the data in along-track using a windowed sinc interpolation, (2) fitting lines to the resampled trajectory with the length of the

SAR aperture and (3) correcting any flight path deviations from the straight lines with phase shifts in the frequency domain.

After the along-track focusing, the channels were combined to increase the SNR and reduce the impact of surface clutter and off-nadir reflections. The delay-and-sum method allows for steering the antenna array beam. The antenna array beam is steered toward nadir by coherently summing the data from each channel while accounting for the actual position of each measurement phase centre. Eleven along-track averages (multilooking) are then performed to reduce speckle in the imagery.

Finally, the different waveform images were vertically combined to increase the dynamic range of the result. The TWT at which the radargrams are combined were chosen with regard to the pulse durations of the transmitted waveforms and the surface return in order to avoid saturation of the high gain channels due to the strong surface return. For 3 waveform collection with $1\mu s$, $3\mu s$, $10\mu s$ nadir waveforms, image 2 is combined with image 1 after 3e-6 s after the surface reflection and image 3 with image 2 10e-6 s after the surface return (see Figure 5).

### 3.3 Resolution and uncertainty analysis

#### 3.3.1 Range resolution

The theoretical range resolution after pulse compression is

$$\delta r = \frac{kc}{2B\sqrt{\epsilon_r}}, \tag{1}$$

where $c$ denotes the speed-of-light in a vacuum, $\epsilon_r$ the real part of the ordinary relative permittivity, $B$ the bandwidth of the

transmitted chirp and $k$ the windowing factor due to the frequency and time domain windows. For the bandwidth of 30 MHz, the theoretical range resolution in ice with $\epsilon_r = 3.15$ and $k = 1.53$ is 4.31 m.





In addition, to estimate the accuracy of a specific target (internal layer or bed return), we have to consider the RMS error of the dielectric constant (CReSIS, 2020a). Here we depend on the exact detection of the ice surface reflection, which is well constrained for our data.

To determine range resolution variability for the bed reflection for the radar data, we performed a crossover analysis of bed pick intersections (see Franke et al., 2020) and calculated the mean deviation $h_c$. We consider an error on the order of 1 % for the dielectric constant for typical dry ice (Bohleber et al., 2012),

$$\sigma_r = \sqrt{(h_c)^2 + \left(\frac{T}{2} 0.01\right)^2},\qquad(2)$$

with the ice thickness $T$ and a mean value for crossover deviation, $h_c$. The full analysis of the range resolution of the bed reflection is documented in Franke et al. (2020) and shows a variability from 13 to 17 m.

### 3.3.2 Bed return resolution

A further parameter is the size of the area illuminated by the radar wave in the bed reflection signal. Here we consider a cross-track resolution for a typical rough surface $\sigma_y$. It is constrained by the pulse-limited footprint and depends on the Tukey and Hanning window parameters,

$$\sigma_y = 2\sqrt{\frac{\frac{H}{\sqrt{\varepsilon_r}}ck_i}{B}},\qquad(3)$$

where $H$ is the elevation of the aircraft over the ice surface. All EGRIP-NOR-2018 flights were performed at an aircraft elevation of $\sim 365$ m above ground. When off-nadir clutter is visible in the radargrams, the cross-track resolution depends on the full beam width, $\beta_y$, of the antenna array,

$$\beta_y = \arcsin\frac{\lambda_c}{Nd_y},\qquad(4)$$

where $\lambda_c$ is the wavelength at the centre frequency, $N$ is the number of array elements and $d_y$ the element spacing of the antennas. The cross-track resolution is now defined as

$$\sigma_y = 2\frac{H+T}{\sqrt{\varepsilon}} \tan\frac{\beta_y k_t}{2}.\qquad(5)$$

For areas without signal-layover, the cross-track footprint ranges between 300–350 m for an ice column of 2000–3000 m in our survey region (Franke et al., 2020). Where layover occurs, we have to consider Equation (5) with a beam angle of $\sim 21°$. Here the cross-track resolution ranges between 800 m to 1100 m.



**Table 2.** Data record properties

| Parameter | Value |
|---|---|
| Covered area | 25 000 km$^2$ |
| Total profile distance | 7494 km |
| Along-track resolution | ~ 27 - 30 m $^{(qlook)}$ / ~ 15 m $^{(SAR_1)}$ / ~ 3 m $^{(SAR_2)}$ |
| Range resolution | 4.31 m |
| Data amount | 4.5 GB $^{(qlook)}$ / 9.5 GB $^{(SAR_1)}$ / 25.6 GB $^{(SAR_2)}$ |

## 4 Results

The design of the EGRIP-NOR-2018 radar survey enables a detailed analysis of the bed and englacial stratigraphy: (i) along radar profiles which are parallel to the ice flow (Figure 1c) and (ii) along radar profiles perpendicular to the ice flow, crossing the shear margins (Figure 1b). In addition, several radar lines cover the location where the northern shear margin shifts its location

(Figure 1a) and an area South-West of EastGRIP where we observe patterns of strong internal ice deformation (location indicated in Figure 1a).

In flow-parallel profiles, we observe three regimes showing different characteristics in the radar stratigraphy. Figure 2 shows a ~200 km long flow-parallel radargram extending from the far upstream end outside of the NEGIS up to 30 km downstream of EastGRIP. The area outside of the ice stream in a distance of 0–60 km shows slightly folded internal layers, which have no

connection to the local basal topography. Steeply dipping internal layers are characterised by a decrease in reflectivity with depth. The part of the profile, which is crossing the shear zone (in 60–90 km distance) is characterised by tight folds in the internal layers, which extend almost up to the ice surface. The folds' apparent wavelength in profile direction depends on the intersection angle with respect to the shear margin orientation. Folds are tightest in profiles oriented ~90° to the shear margin fold axis. We observe the best resolution of internal layers inside the ice stream, particulary in the lower part of the ice column.

All distortions in the internal stratigraphy inside the ice stream seem to be related to the underlying bed.

Radar profiles oriented perpendicular to ice flow show a strong imprint of NEGIS' dynamics. The most striking features are tight folds in the area of the shear margins. Bright stripes characterise them in the radargrams (e.g. Figure 3), which represent a loss in return power due to steeply inclined internal reflectors (Holschuh et al., 2014). The onset of this kind of folding starts at the shear zone's outer boundary (marked by yellow triangles in Figure 3c and d). Depending on the ice stream's location

and width, these folds can be traced towards the ice stream centre for up to tens of kilometres, also towards locations where no shearing at the ice surface is observed. This becomes evident by comparing the upstream and further downstream radargram in Figure 3c and d.

At several locations in flow-perpendicular radargrams we observe a drawdown of the radar stratigraphy towards the shear zone's outer margin (see Figure 3c). In general, the stratigraphy of internal layers North-West of the NEGIS differs from the

stratigraphy South-East of the NEGIS. The North-West is much more undisturbed than the South-East. A distinct example is shown in Figure 3a and b. The stratigraphy is marked by long-wavelength anticlines and synclines with elevation differences





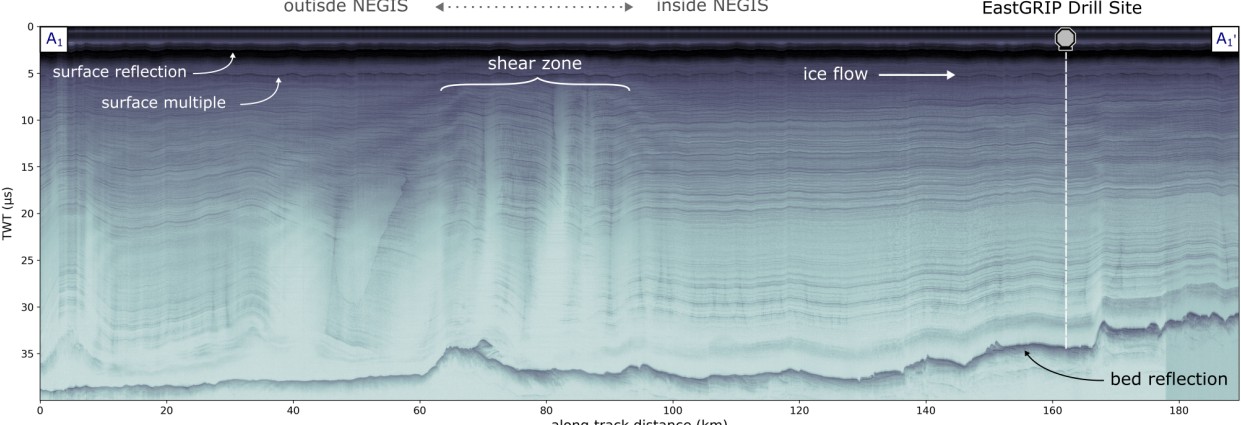

**Figure 2.** Flow-parallel radar profile *A* composed of three frames: 20180512_01_001-004. The location is indicated in Figure 1a and covers the area outside of the ice stream, the shear zone and the ice stream's trunk. The position of the EastGRIP drill site is labeled and indicated with a white dashed vertical line.

of almost half of the ice column. In the anticlines' cores, we find strong englacial reflections, which have been misinterpreted before as bedrock (Franke et al., 2020). We note that some of the englacial reflections appear to be attached to the basal reflection (Figure 3a). Figure 3b shows that the deformations patterns in the anticline cores are very complex.

## 4.1 Data products


We offer three different data products of the EGRIP-NOR-2018 radar survey: (i) quick-looks (qlook), (ii) SAR focused ($SAR_1$) and (iii) SAR focused with a large aperture ($SAR_2$). The general data record properties are shown in Table 2 and detailed documentation can be found in the CReSIS MCoRDS documentation (https://data.cresis.ku.edu/) and on the CReSIS Wiki Website (https://ops.cresis.ku.edu/wiki/index.php/Main_Page). In Figure 4 we provide an overview of the differences between

these three data products of the radargrams.

### 4.1.1 qlook

This product uses unfocused synthetic aperture radar processing for each channel and assumes that all reflections arrive at the receiver from nadir. The data are coherently stacked in slow time and no correction for propagation delay changes is applied. Here, no motion compensation is applied. Finally, the signals from all eight channels are averaged incoherently. The range

resolution is the same as for all other products. The trace spacing is ~27–30 m.

### 4.1.2 SAR with default settings ($SAR_1$)

This data product uses focused synthetic aperture radar processing (fk migration) on each channel individually. The SAR processing requires a uniformly sampled linear trajectory along the extent of the SAR aperture. Motion compensation is applied





**Figure 3.** Radar profiles $C_{1,2,3}$ (frames 20180508_06_003, 20180511_01_007 and 20180514_01_011-012, respectively). The corresponding locations are shown in Figure 1a. Panel (a) shows a profile located outside of the NEGIS, SE of EastGRIP. A close up of an anticline and other patterns of deformation are shown in panel (b), resembling the skeleton of a dead penguin. A slice through the ice stream in the upstream region is shown in panel (c) and a radar section further downstream in (d). The position of the shear margins (the maximum in the surface velocity gradient) are indicated with a yellow triangle and the folds in the shear margin areas with purple lines. The ice flow direction is out of the page.

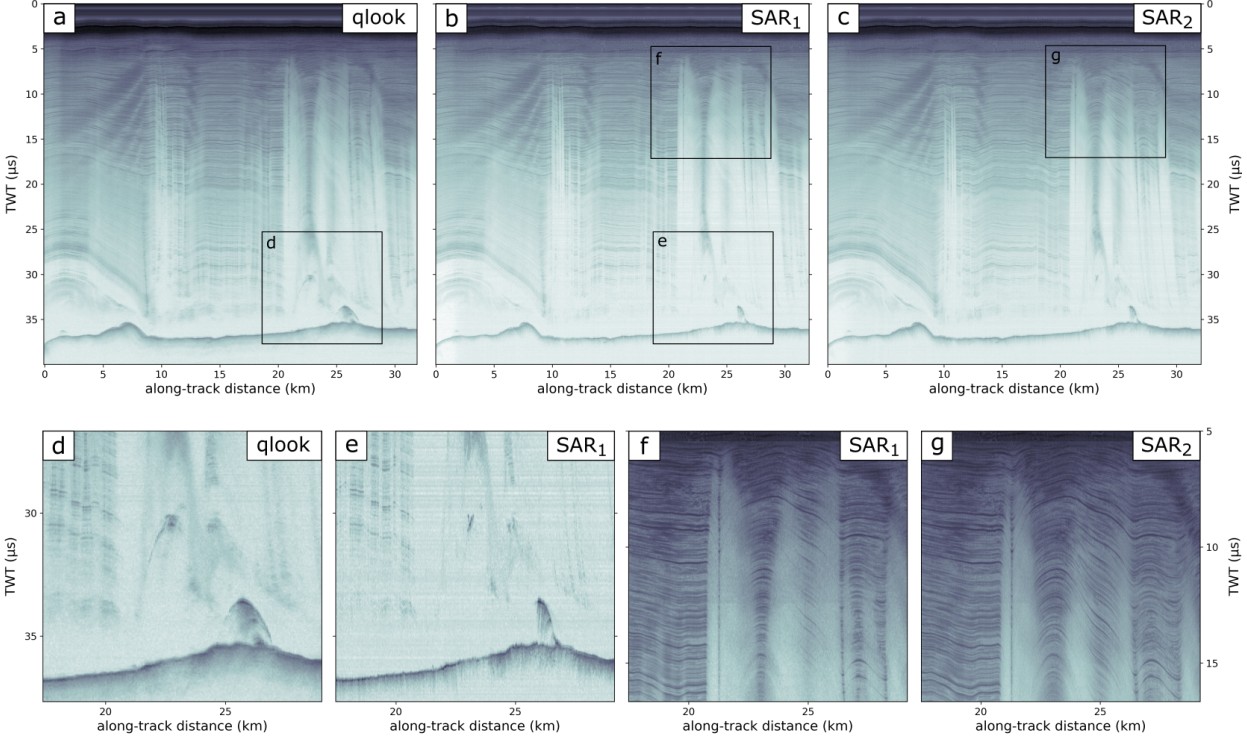

**Figure 4.** Radargrams of a flow-perpendicular radar profile showing the three different data products: (a) quick-look (qlook) processed, (b) SAR focused (SAR$_1$) and (c) SAR focused with a larger aperture (SAR$_2$). The main difference between qlook and SAR$_1$ is the focusing of signals via fk migration (panel d and e). In contrast do the default SAR focusing (panel f) a larger aperture enables a better resolution of steeply inclined internal layers (panel g).

using high precision processed GPS and INS data from the aircraft. The direction of arrival is estimated by delay-and-sum

beamforming to combine the channels. A Hanning window is applied in the frequency domain to suppress sidelobes. This product is comparable to the CReSIS standard data product. The trace spacing is ~15 m.

### 4.1.3 SAR with wider angular range (SAR$_2$)

By processing at a finer SAR resolution, the SAR processor uses scattered energy from a wider angular range around nadir to form the image. Since the angle of scattered returns from a specular internal layer is proportional to the internal layer slope,

the SAR processor's increased sensitivity to larger angle returns translates to an increased sensitivity to layers with larger slopes. We achieve a better resolution of steeply inclined internal reflectors by changing the along-track resolution before SAR processing to 1 m ($\approx \sigma_x = 1$, whereas the default setting is $\sigma_x = 2.5$). 1 m is not the smallest possible value for processing, but is on the limit to achieve a sufficiently high SNR. The SNR is smaller for larger angles because the range to the target increases



for greater angles, which leads to additional signal loss (spherical spreading loss and additional signal attenuation in ice). The

differences between radargrams processed with $\sigma_x = 1$ and 2.5 are shown in Figure 4. The final trace spacing is ~3 m.

### 4.1.4   Individual waveforms

The combination of the three waveform images will increase the dynamic range of the whole radargram. However, specific analyses may require only a single waveform. Therefore, we provide the respective echogram data for each waveform separately (see Figure 5). The files are labelled with *img_01*, *img_02* and *img_03* for image 1, 2 and 3, respectively.

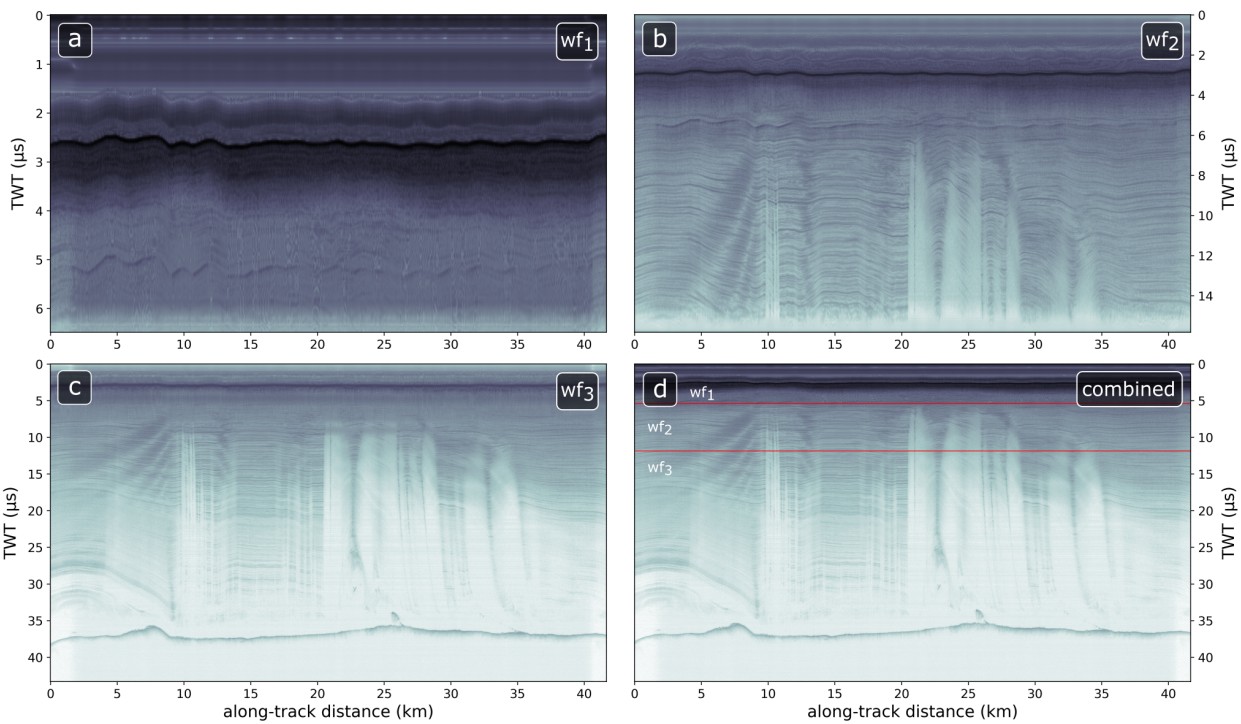

**Figure 5.** Radargrams of the frame 20180517_01_008 subdivided into (a) waveform 1, (b) waveform 2, (c) waveform 3 and (d) the radargram composed of all three waveforms in combination. The locations, where the different waveforms are concatenated, are indicated with red lines in (d). Waveform 1 and 2 are plotted at full range. Note the different resolution of the near-surface stratigraphy in waveforms 1 to 3.

### 245   4.1.5   Image mode flights

In addition to the data recorded in the so-called sounding mode, the data set also contains two segments (20180510_02 and 20180515_01) recorded in the image mode (see Table A1). The acquisition settings of these segments are slightly different since the transmission signals are composed of four instead of three pulses: 1 $\mu$s and 3 $\mu$s waveforms with nadir directed transmit beam followed by two 10 $\mu$s waveforms, one with the transmit beam directed to the left and one with it directed to the

right to increase the imaged swath width at the ice bottom. We processed the data in a way that the 10 $\mu$s left and 10 $\mu$s right





signal are steered towards nadir during data processing. This is possible because both waveforms contain nadir information since the beams overlap at nadir. However, the reflection power from nadir is reduced because the two side looking transmit beams have reduced gain by about 3-4 dB relative to when the beam is pointed directly at nadir. The third waveform (see Figure 5 c and d) in the combined image for the two particular segments has been computed from the coherent combination of both, the 10 $\mu$s left and 10 $\mu$s right return signals.

## 4.2 Data formats

We provide the main radar data and auxiliary data in the following formats:

1. The radar data containing a matrix of the echogram and the corresponding GPS information, such as coordinates, aircraft elevation, and timing of every trace, are stored as *matfiles* (HDF-5 based format). The echograms are provided for the combined waveform product as well as for the individual waveforms. Furthermore, these files contain cell arrays with information about all processing parameters used.

2. A set of figures for each profile, showing the radargram and its respective location in the EGRIP-NOR-2018 survey.

3. A set of shapefiles (lines) containing the location of every frame.

4. An Excel Spreadsheet which contains all parameter settings applied during radar data processing.

In the Appendix we describe the data architecture and how the data is stored in the PANGAEA repository.

## 5 Relevance of the data set

This radar data set provides essential observations of internal and bed reflections to determine spatial distribution of ice thickness, internal layering and reflectivity. These observables constitute boundary conditions and elucidate properties and processes of the NEGIS. The data comprises ~7500 km high-resolution radar data to the scientific community. In contrast to previous surveys in this area, the data presented here were specifically recorded over a broad spatial extent to understand the history of the NEGIS and our general understanding about ice streams. The tightly spaced radar profiles perpendicular to ice flow allow a 3D interpretation of the ice-internal stratigraphic architecture. Because most radar lines are directly or indirectly spatially connected to the location of the drill site of the EastGRIP ice core, these data are significant for various objectives regarding the ice-dynamic understanding of NEGIS as well as interpretation of the climate proxy record retrievable from the ice core. With this data set the scientific community will be able to upscale the findings of the EastGRIP project from the location of the ice core to the immediate surrounding of the upstream part of the NEGIS. The prospect that parts of the ice core can be rotated back into their correct geographic direction (Westhoff et al., 2020) also allows a systematic analysis of ice crystal orientation fabric together with the radar data. Furthermore, the data presented here can be combined with the radar data acquired during Operation Ice Bridge (OIB) to extend the large-scale understanding of glaciological properties in the Greenland Ice Sheet.





## 6  Conclusions

We present a high-resolution ice-penetrating radar data set at the onset region of the NEGIS. The EGRIP-NOR-2018 radar data reveal the internal stratigraphy and bed topography of the upstream part of the NEGIS in high vertical and horizontal resolution, given the dense coverage. Several survey lines intersect at the EastGRIP drill site location, enabling a combination of both data sets. Ultimately, this data set will improve our understanding of the NEGIS in its present form and also contributes to our understanding of its genesis and evolution. Radar and auxiliary data will be provided as *matfiles* for the combined echograms as well as for the individual waveforms. The radar data products comprise unfocused data (qlook), SAR focused data ($SAR_1$) and SAR focused data with a wider angular range ($SAR_2$)

## 7  Code and data availability

The EGRIP-NOR-2018 radar data products are available at the PANGAEA Data Publisher (https://doi.pangaea.de/10.1594/PANGAEA.928569; Franke et al. 2021a). The EGRIP-NOR-2018 bed topography (Franke et al. (2019) is available under https://doi.pangaea.de/10.1594/PANGAEA.907918). The CReSIS-toolbox is available under https://github.com/CReSIS/ and the main documentation can be found at https://ops.cresis.ku.edu/wiki/. The MEaSUREs Greenland Ice Sheet Velocity Map from InSAR Data, Version 2 from Joughin et al. (2017) is available from https://doi.org/10.5067/OC7B04ZM9G6Q.



## Appendix A: Additional Information for the Segments

The data are stored in *zip* archives for each segment and processing product, respectively. For details on the specifications of each segment and their respective coverage, see Table A1 and Figure A1. The filenames of the archives are composed of the segment and the data product (e.g. 20180508_02_qlook.zip, 20180508_02_sar1.zip and 20180508_02_sar2.zip for the quick-look, SAR with default settings ($SAR_1$) and SAR with larger angular range data product ($SAR_2$), respectively). Each *zip* archive contains the individual frames in the *matfile* format.

**Table A1.**

| Segment | Frames | Frequency Range | Waveforms (pulse length and direction) | Segment Length |
|---|---|---|---|---|
| 20180508_02 | 2 | 180–210 MHz | 3 ($1\mu s$, $3\mu s$, $10\mu s$ nadir) | 72 km |
| 20180508_06 | 4 | 180–210 MHz | 3 ($1\mu s$, $3\mu s$, $10\mu s$ nadir) | 189 km |
| 20180509_01 | 18 | 180–210 MHz | 3 ($1\mu s$, $3\mu s$, $10\mu s$ nadir) | 852 km |
| 20180510_01 | 15 | 180–210 MHz | 3 ($1\mu s$, $3\mu s$, $10\mu s$ nadir) | 726 km |
| 20180510_02 [a] | 15 | 180–210 MHz | 4 ($1\mu s$ and $3\mu s$ nadir, $10\mu s$ left, $10\mu s$ right) | 675 km |
| 20180511_01 | 13 | 180–210 MHz | 3 ($1\mu s$, $3\mu s$, $10\mu s$ nadir) | 721 km |
| 20180512_01 | 15 | 180–210 MHz | 3 ($1\mu s$, $3\mu s$, $10\mu s$ nadir) | 635 km |
| 20180512_02 | 14 | 180–210 MHz | 3 ($1\mu s$, $3\mu s$, $10\mu s$ nadir) | 645 km |
| 20180514_01 | 19 | 180–210 MHz | 3 ($1\mu s$, $3\mu s$, $10\mu s$ nadir) | 750 km |
| 20180514_03 | 12 | 180–210 MHz | 3 ($1\mu s$, $3\mu s$, $10\mu s$ nadir) | 740 km |
| 20180515_01 [a] | 16 | 180–210 MHz | 4 ($1\mu s$ and $3\mu s$ nadir, $10\mu s$ left, $10\mu s$ right) | 733 km |
| 20180517_01 | 20 | 180–210 MHz | 3 ($1\mu s$, $3\mu s$, $10\mu s$ nadir) | 753 km |

[a]The nadir part of all four waveforms was used for image combination



**Figure A1.** Radar profiles locations of the 12 segments shown in Table A1 of the EGRIP-NOR-2018 data set. The respective segments are highlighted with a white line and the radar profiles of the complete survey with a finer black line. The background map represents ice surface velocity from Joughin et al. (2017). The shear margin is indicated with a white dashed outline and the location of EastGRIP with a red dot.



*Author contributions.* Steven Franke, Nils Dörr and Tamara Gerber wrote the manuscript. Tobias Binder, Daniela Jansen and John Paden acquired the radar data in the field. Olaf Eisen and Daniela Jansen were PI and co-PI of the radar campaign. Tobias Binder and Steven Franke processed the radar data with the support of John Paden, Daniela Jansen, Veit Helm and Daniel Steinhage. All authors discussed and revised the manuscript.

*Competing interests.* The authors declare that they have no conflict of interest.

*Acknowledgements.* We thank the AWI and Ken Borek crew of the research aircraft Polar6. Logistical support in the field was provided by the East Greenland Ice-Core Project. EastGRIP is directed and organized by the centre of Ice and Climate at the Niels Bohr Institute. It is supported by funding agencies and institutions in Denmark (A. P. Møller Foundation, University of Copenhagen), USA (U.S. National Science Foundation, Office of Polar Programs), Germany (Alfred Wegener Institute, Helmholtz Centre for Polar and Marine Research), Japan (National Institute of Polar Research and Arctic Challenge for Sustainability), Norway (University of Bergen and Bergen Research Foundation), Switzerland (Swiss National Science Foundation), France (French Polar Institute Paul-Emile Victor, Institute for Geosciences and Environmental research) and China (Chinese Academy of Sciences and Beijing Normal University). We acknowledge the use of the CReSIS toolbox from CReSIS generated with support from the University of Kansas, NASA Operation IceBridge grant NNX16AH54G, and NSF grants ACI-1443054, OPP-1739003, and IIS-1838230.



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
