# Peer review of "Airborne ultra-wideband radar sounding over the shear margins and along flow lines at the onset region of the Northeast Greenland Ice Stream"

_Earth System Science Data, 2021_

## Author Comment (AC1)

Color scheme for the responses:

Text of the reviewer in black .
Response of the authors in **blue**.
Removed text/figure in the manuscript in **red**.
Added text/figure in the manuscript in **green**.

Review of "Airborne ultra-wideband radar sounding over the shear margins and along flow lines at the onset region of the Northeast Greenland Ice Stream" by Franke et al..

This manuscript describes a newly constructed airborne radar sounding dataset over the initiation zone of Greenland's largest outlet glacier, NEGIS. The survey area covers several hundred kilometers along strike, including the EastGRIP drill site, and with many transects perpendicular to ice flow. The authors describe the data collection and processing steps, resulting in the construction of numerous 'cleaned' radargram profiles for download by interested readers.

The authors have written a concise and well formated manuscript which achieves their goals of presenting and releasing a new radar dataset of NEGIS. This dataset covers a larger spatial extent than previous products, and has been robustly processed and archived, and hence is a valuable dataset for future work. I only have minor comments, as the paper is already of high quality, and is suitable for publication in ESSD.

**Response:**

We would like to thank the reviewer for his constructive and positive evaluation of our manuscript. The answers to the suggestions and comments of the reviewer can be found in the further course of the text.
* * *
Minor comments:

1). The paper could be strengthened by emphasazing with more precise examples what types of new insights, and scientific discoveries, this dataset may help address. In the `Relavance of the data set' section, they say "These observables constitute boundary conditions and elucidate properties and processes of NEGIS" (Lines, 268 - 269), which is true, but it is vauge. Perhaps the authors should identify the key things in glaciology that refined bed and ice column images could help reconcile: e.g., basal hyrdaulic processes, stick-slip processes, rheologic deformation laws (Podolskiy and Walter, 2016), etc.

**Response:**

We agree with the reviewer that more specific examples on which aspects in glaciology can be investigated with the data. We therefore added the following to the section "Relevance of the data set":

The tightly spaced survey lines allow to derive the information on the past and present ice flow dynamics revealing the paleo-processes of the NEGIS, in particular when combined with ice core data. Hereby it might be possible to address the question of how long the NEGIS has been active in its present form. The radar data furthermore allows a systematic analysis of the return power of internal layers and the bed and could potentially reveal information about the englacial temperature regimes, properties of the bed and basal hydrology (e.g. Franke et al., 2020, 2021b) and provide crucial boundary conditions for ice flow models. In addition our data holds the potential of mapping the crystal orientation fabric (COF) and especially the horizontal anisotropy from the birefringence effects (e.g. at the shear margins or at intersecting radar profiles of different polarization directions. COF is another important, but yet poorly constrained parameter in ice flow models.

The connections of their data to the many previous studies at NEGIS could also be highlighted more; such as, are the main features of these data consistent with Christianson et al., 2014, Villelonga et al., 2014, Riverman, 2019, etc?

**Response:**

We agree and like the idea to connect the findings which have been achieved with this data set to the ones in the studies mentioned here. However, this would be some sort of a summary citing the literature, which has been analyzing and interpreting the data (e.g. the results of Franke et al. (2020, 2021), because this manuscript itself is not supposed to perform any analysis or interpretation, according to ESSD guidelines.

- **Christianson et al. (2014):** A statement that the key findings of Franke et al. (2021; who use the EGRIP-NOR-2018 data) are in agreement with the results of Christianson et al. (2014) is already made in the section "Study area and previous work".

- **Riverman et al. (2019)**: We added the following statement to second to last paragraph of the Results section:

  The drawdown could be explained by temperate ice at the base (Franke et al., 2021c). Melting at the base at this location would be in agreement with subglacial hydrology modeling results by (Riverman et al.,2019a), who suggest that melt out of sediments within the ice column potentially creates the subglacial bedforms they identified.

- **Vallelonga et al. (2014):** We actually believe that a specific comparison of the radar data in this study with the results of Vallelonga et al. (2014) is not necessary because Vallelonga et al. (2014) take all results and interpretations from Christianson et al. (2014) as they state in their manuscript: "*Full analysis of the basal interface using RES and active-source seismic data is detailed in Christianson et al. (2014).*"

2). In  Figure 3), caption, it's stated, "ice flow direction is out of page". However, the ice flow direction is actually "into the page", as can be seen since c1', c2', c3', are all on the right hand side of these transects, and in map view Fig. 1, the c1', c2', c3', are on the south-east side of the ice stream, and of course NEGIS flows north.

**Response:**

The reviewer is completely right. We corrected this and now state that ice flow is into the page.
* * *
3). In discussing the flow perpendicular radargrams, they state " In the anticlines' cores, we find strong englacial reflections, which have been misinterpreted before as bedrock (Franke et al., 2020)". (Lines 212 - 213). Can the authors clarify this point? What new lines of evidence are being used here, compared with the study Franke et al., 2020, to re-interpret these features?

**Response:**

In respect to the statement that we find "strong englacial reflections in the anticlines' cores, which have been misinterpreted as bedrock before", we are not making a re-interpretation of the features. We just describe the radargram and what is already published by Franke et al., (2020), which is also the reason why we cite this paper.

The remaining sentences of this paragraph, "We note that some of the englacial reflections appear to be attached to the basal reflection (Figure 3a). Figure 3b shows that the deformations patterns in the anticline cores are very complex" (Lines 213 - 214) could also be expanded to include more details.

**Response:**

We agree. We added the following statement after the first sentence:

This could be an indication that the strong englacial reflection could be basal material (e.g. soft sediments) that were transported upwards by the folding.

After the second sentence (L213-214), we added:

Some of the reflection patterns (e.g. in Figure 3b) result in connected structures similar to the isochrones in the upper part of the ice column. In addition, we find isolated patches with high reflectivity, which, however, do not show in a coherent pattern (highlighted in Figure 3b).

Moreover, we added some labels in Figure 3b.
* * *
4). In relation to SAR processing, it's stated that "We used a two-layered velocity model with constant permittivity values for air (r = 1) and ice (r = 3.15)" (Line 137).  Can the authors clarify that no problems are encountered by ignoring the uppermost ~50 - 100 m of firn in the

ice? Clearly the ice column is not precisely a homogenous medium, so the authors should qualitatively (or quantitaively) justify the effects of this approximation.

**Response:**

The reviewer is right, the ice column is not a homogeneous medium. This accounts for the vertical component (ice column depth) as well as for the horizontal component. In respect to the latter, the effect of firn on the EM wave propagation velocity will vary significantly in this highly dynamic region. That the firn is much more compacted at the shear margins as compared to the area outside and in the centre of the ice stream has been shown by Riverman et al. (2019). This point has been also raised by Franke et al. (2020) for the publication of the bed topography, where the authors do specifically exclude a firn correction because they consider constant value not to be an appropriate correction for this specific region.

We totally agree that there is an uncertainty for the SAR processing due to the firn layer. However, we argue that for the SAR processing the uncertainties of the firn layer in EM wave propagation velocity is very small and should not affect the fk-migration in the SAR processor. Moreover, a constant value for $\varepsilon_{ice}$=3.15 is the standard also for all publicly available OIR data.

In the manuscript we added therefore the following sentence:

We consider the constant value of $\varepsilon_{ice}$=3.15 to be justified, since the uncertainties in the propagation velocity in the firn layer have a negligible effect on the fk-migration.
* * *
Additional References:

**Riverman, K. L.**, Alley, R. B., Anandakrishnan, S., Christianson, K., Holschuh, N. D., Medley, B., Muto, A., and Peters, L. E.: Enhanced FirnDensification in High-Accumulation Shear Margins of the NE Greenland Ice Stream, Journal of Geophysical Research: Earth Surface,124, 365–382, https://doi.org/10.1029/2017JF004604, 2019

**Franke, S.**, Jansen, D., Binder, T., Dörr, N., Helm, V., Paden, J., Steinhage, D., and Eisen, O.: Bed topography and subglacial landforms in the onset region of the Northeast Greenland Ice Stream, Annals of Glaciology, 61, 143–153, https://doi.org/10.1017/aog.2020.12, 2020

**Franke, S.**, Jansen, D., Beyer, S., Neckel, N., Binder, T., Paden, J., and Eisen, O.: Complex Basal Conditions and Their Influence onIce Flow at the Onset of the Northeast Greenland Ice Stream, Journal of Geophysical Research: Earth Surface, 126, e2020JF005 689,https://doi.org/10.1029/2020JF005689, https://agupubs.onlinelibrary.wiley.com/doi/abs/10.1029/2020JF005689, 2021.

---

## Author Comment (AC2)

Color scheme for the responses:

Text of the reviewer in **black**.
Response of the authors in **blue**.
Removed text/figure in the manuscript in **red**.
Added text/figure in the manuscript in **green**.

This manuscript presents a high-resolution airborne radar data set (EGRIP-NOR-2018) for the onset region of NEGIS. We found that the authors have used this data set to produce and publish the gridded ice thickness and bed topography data as well as the TWTs of the ice thickness along the radar profiles in the onset region of NEGIS, which constitute important boundary conditions for numerical model. Even so, the data presented in this manuscript is of exciting importance. From it, we can also derive the characteristics of isochronous layer to reveal the historical properties and processes of the NEGIS, especially when combined with ice core data. There are, however, major issues with the manuscript that would be valuable to address.

**Response:**

We would like to thank the reviewer for his constructive and generally positive evaluation of our manuscript. The answers to the suggestions and comments of the reviewer can be found in the further course of the text.
* * *
First, the data set is not accessible via the given identifier in the paper (https://doi.pangaea.de/10.1594/PANGAEA.928569) may for it is still under review (it shows "The rights given by your login do not allow downloading of dataset #928569. Please login with another user name!"). As a result, I have not been able to assess whether the data set meets the requirements of the journal.

**Response:**

We are sorry that the reviewer did not find access to the data. It is correct that the given PANGAEA link does not allow access (yet) to the data and is locked until the manuscript is published. We did this because of the open review process of ESSD and otherwise everyone would have been able to download the data prior to the peer review of the manuscript.

We created a temporary link with full access at the initial submission of the manuscript, which was sent to the editor to forward the link to the reviewers (as obviously the authors are not supposed to communicate with the referees directly). We have been in contact with the topical editor and editorial support that this link will be sent to the reviewer, to access and check the data.
* * *
Second, "line 63" says "unique airborne radar data". I think it is a needless over-assertion which weakens the credibility of the authors and manuscript. Does IceBridge have any observations in this area? If so, I think the authors should make a cross-comparison with IceBridge to validate the data set. If not, can use crossover analysis to validate the data set. In addition, can the authors give the calculation and accuracy of GPS and INS in this manuscript? From the manuscript, they have an important impact on the accuracy of the data set. In a word, I think the accuracy evaluation of the data set is not enough.

**Response:**

**L63**
We removed the term "unique".

**Operation IceBridge (OIB) data**
We appreciate the idea of the reviewer. A cross-comparison of CReSIS/OIB data would be a feasible idea to validate and compare the data sets. However, we are not completely sure what the reviewer is suggesting with a crossover analysis to validate the data set. If the reviewer is referring to a crossover analysis of ice thickness, we refer to an earlier publication (Franke et al., 2020) where ice thickness and bedrock topography has been compared with other available products. However, ice thickness data is not part of this data description manuscript, but the radar data as such. We believe that a robust crossover of ice thickness of our data set and OIB data (acquired over multiple decades) would include several steps to assure that the data has been processed and the surface and bed reflection determined in a consistent way.

Nevertheless, we included a comparison of selected OIB and AWI radargrams, which are located closely to each other or intersect. We expand the Appendix section of our manuscript and include a few figures where we compare OIB radargrams with our data. Nevertheless, we find this of little added value, as the hardware as well as the processing software of the AWI UWB and OIB MCORDS systems are basically identical, apart from being different versions.

We added the following text and figure in the Appendix (B):

**Appendix B: Comparison to OIB surveys**

We evaluate the quality of the EGRIP-NOR-2018 radar data by comparing selected profiles with OIB radargrams. Figure B1 a shows two locations in our survey regions, where we compare two intersecting radargrams, respectively. For the comparison, we focus on the environment outside the ice stream in the southeast, where we observe large englacial folds (Figure B1 b), and the radar stratigraphy along the shear zones (Figure B1 c,d). In summary, the comparison shows that the EGRIP-NOR-2018 data have a comparable quality and resolution of the internal layers as well as the bed reflection. In Figure B1 c and d we note that the steeply dipping internal layers are slightly better resolved in our data set.

[Figure]

**Figure B1.** Comparison of selected AWI UWB radargrams with OIB radargrams. (a) 3D view on the EGRIP-NOR-2018 survey highlighting the location of: (b) two intersecting radargrams (OIB profile 20130402_01_026 and AWI profile 20180512_02_003), and (c,d) two nearly parallel oriented profiles showing both shear margins of NEGIS (OIB profile 20070912_01_005 and AWI profile 20180514_01_015).

**GPS/INS**

Regarding the calculation and accuracy of GPS and INS data we added the following paragraph (and restructured the text to make it consistent) to improve the accuracy evaluation of our data:

To estimate the flight trajectory we used Novatel OEM6 receivers at 20Hz data rate. The precise point positioning (PPP) post processed accuracy (commercial software package Waypoint 8.4) is estimated to be better than 3 cm for latitude and longitude and better than 10 cm for altitude. INS data was acquired by the onboard laser gyro inertial navigation system (Honeywell LASERREF V). Its accuracy is given to be better than 0.1° for Pitch and Roll and better than 0.4° for True Heading (Honeywell Product description).

I made some specific comments and suggestions below, which I hope will help improve this paper.

L175: What does "ki" mean in equation 4?

L182: What does "kt" mean in equation 5?

**Response:**

We are glad that the reviewer found both mistakes in L175 and L182 in equations 4 and 5. We actually changed the nomenclature for the "k"s in the equations in the following way:

**eq1**: k is now $k_t$ and describes the windowing factor due to the frequency and time domain windows

**eq4**: ki is now $k_t$ because it is the same constant as in eq1.

**eq5**: we changed $k_t$ to $k_y$.
$k_y$ is the approximate cross-track windowing factor for a hanning window applied to a small cross-track antenna array.

The explanation for the constants is now added in the text below eq1 and eq5.
* * *
Figure 3 "C1,2,3" should be "C1-C1', C2-C2', C3-C3'", keep the same with Figure 1

**Response:**

We believe this suggestion is referring to the beginning of the caption in Figure 3. Thus, we changed the following in the caption:

$C_{1,2,3} \rightarrow C_1\text{-}C_1', C_2\text{-}C_2', C_3\text{-}C_3',$